# A UNIFIED THEORY OF SCENE REPRESENTATION LEARNING AND OBJECT REPRESENTATION LEARNING

## ABSTRACT

The goal of representation learning is the unsupervised learning of simple and useful representations that model sensory input. Various methods have been proposed in representation learning, but a unified theory has not yet been established. Two problems exist in the representation learning of a visual scene that contains multiple objects: scene representation learning and object representation learning. Scene representation refers to decomposing a single visual scene that contains multiple objects into a combination of multiple individual objects. Object representation refers to decomposing a single object into a combination of multiple attributes, such as position and shape. Scene representation learning and object representation learning have been formulated in different ways in previous studies. Recently, Ohmura et al. (2023) proposed a theory of object representation learning in which transformations between two objects are learned to satisfy algebraic independence so that one attribute of a single object can be transformed while the other remains invariant. In existing methods of object representation learning, independence is often imposed between scalar variables, whereas theory based on algebraic independence successfully weakens the constraint from between scalar variables to between latent vectors. The latent vector is also used to represent an individual object in existing methods of scene representation learning because such a vector can contain more information than the scalar variable. Furthermore, one of the main components of algebraic independence is commutativity. Existing methods of scene representation learning typically represent a visual scene as the sum of multiple object representations, and the sum satisfies commutativity. We focused on the commonalities between object representation learning and scene representation learning: constraints between latent vectors and commutativity. We proposed a unified theory based on algebraic independence that explains both scene representation learning and object representation learning. We validated our theory in experiments on an image dataset that contained multiple objects.

## 1 INTRODUCTION

The goal of representation learning is the unsupervised learning of simple and applicable representations that model sensory input (Bengio et al., 2013). Various representation learning methods have been proposed, such as those based on variational autoencoders (Kingma & Welling, 2014; Higgins et al., 2017), generative adversarial networks (Goodfellow et al., 2014; Chen et al., 2016), Lie group transformations (Takada et al., 2022), and manifolds (Fumero et al., 2021). Several studies have formulated based on group theory toward a unified theory of representation learning (Higgins et al., 2018; Takada et al., 2021); however, such a theory has not yet been established.

Two problems exist in the representation learning of a visual scene that contains multiple objects: scene representation and object representation. Scene representation refers to decomposing a single visual scene that contains multiple objects into a combination of multiple individual objects. Object representation refers to decomposing a single object into a combination of multiple attributes, such as position and shape. A number of previous studies were conducted in which scene and object representations were learned simultaneously (Burgess et al., 2019; Greff et al., 2019; Crawford & Pineau, 2019; Engelcke et al., 2019; 2021; Emami et al., 2021; Lin et al., 2020; Vikström & Ilin, 2022; Jia et al., 2022; Seitzer et al., 2023; Singh et al., 2022; Jiang et al., 2023). For scene rep-

resentation, existing methods typically represent an input image by segmenting it into component images by ensuring that each component image is uniquely bound to each single latent vector. The input image is then modeled as the sum of the component images. For object representation, existing methods typically constrain the latent vectors to satisfy stochastic independence (Higgins et al., 2017). Thus, scene representation learning and object representation learning appear to be based on very different mechanisms.

Recently, Ohmura et al. (2023) proposed a theory of object representation learning in which the transformation between two objects is learned to satisfy algebraic independence so that one attribute of a single object can be transformed while the other remains invariant. The algebraic independence structure in category theory (Simpson, 2018) is a generalization of several independences in mathematics, including orthogonality and stochastic independence used in representation learning (Higgins et al., 2017; Fumero et al., 2021). In conventional methods of object representation learning, independence is often imposed between scalar variables, whereas Ohmura et al. (2023) successfully weakens the constraint from between scalar variables to between latent vectors. By such a relaxation, the object can be represented by decomposing it into attributes, such as color and shape, without separating them into scalar variables such as red and blue. Latent vectors are also used to model individual objects in existing methods of scene representation learning. Furthermore, one of the key components of algebraic independence is commutativity, and the summation used in scene representation also satisfies commutativity. Therefore, based on the commonality of commutativity and the relaxation to independence between vectors, we assume that both object representation learning and scene representation learning can be explained by a unified theory.

In this study, we propose a unified theory based on algebraic independence that explains both object representation learning and scene representation learning. We formulate scene representation learning as satisfying algebraic independence on scene transformations, which refers to the transformation from a visual scene that contains multiple objects to another scene. We also formulate object representation learning as satisfying algebraic independence on object transformations, which refers to the transformation from a single object to another single object. We validate our theory in experiments on an image dataset that contains multiple objects.

Our contributions can be summarized as follows:
- We algebraically formulate scene representation learning, which previous representation learning studies based on algebraic formulation (Higgins et al., 2018; Ohmura et al., 2023) did not address.
- We provide a unified explanation for the learning of scene and object representations, which were formulated differently in previous studies, using algebraic independence.

## 2 UNIFIED THEORY BASED ON ALGEBRAIC INDEPENDENCE

In this section, we describe our unified theory of object representation learning and scene representation learning. Our theory considers the algebraic structure among multiple neural network (NN) models that is common to both object representation learning and scene representation learning. In object representation learning, Ohmura et al. (2023) considered the algebraic structure among multiple NN models and formulated object representation learning as learning transformations between two objects so that they satisfy algebraic independence. As the algebraic structure, we adopt the formulation of Ohmura et al. (2023). Conventional methods for object representation learning have focused on constraints on data distribution, such as stochastic independence (Chen et al., 2016; Higgins et al., 2017). Higgins et al. (2018) formulated object representation learning based on the assumption that world dynamics have an algebraic structure. They considered constraints on the data distribution; however, they did not consider the algebraic structure among multiple NN models.

Existing methods of scene representation learning typically represent single objects using latent vectors and represent a scene (an image that contains multiple objects) by decoding latent vectors. Studies have been conducted that consider the algebraic structure in object representation learning (Higgins et al., 2018; Ohmura et al., 2023); however, no such studies have been conducted in scene representation learning, and the connection between object representation learning and scene representation learning has not been clear. In this study, we describe object representation learning and scene representation learning in a common framework for the first time. Thus, we formulate scene representation, in addition to object representation, as transformations between scenes. Then we generalize the existing method of scene representation learning in terms of the required alge-

braic constraints between transformations. As a result, we find that algebraic constraints for scene representation learning also require algebraic independence.

## 2.1 OBJECT REPRESENTATION

We briefly describe object representation learning in the study by Ohmura et al. (2023). Object representation is described through a relationship between two single objects X and Y. The relationship between X and Y is formulated using $N$ transformations as follows:

$$\mathrm{Y} = F_0(\boldsymbol{\lambda}_0)F_1(\boldsymbol{\lambda}_1)...F_{N-1}(\boldsymbol{\lambda}_{N-1})[\mathrm{X}]. \tag{1}$$

$F_0, ..., F_{N-1}$ are transformations (functions whose input and output dimensions are the same) and are learned using NN models. $\boldsymbol{\lambda}_0, ..., \boldsymbol{\lambda}_{N-1}$ are transformation parameters and vectors.

Object representation learning is defined as learning so that the transformations $F_0, ..., F_{N-1}$ are algebraically independent. Algebraic independence consists of three conditions as follows:

**(1) Commutativity**   The result of transformations from X is the same Y regardless of the order of the transformations $F_0, ..., F_{N-1}$. For $N = 2$, for example, $\mathrm{Y} = F_0(\boldsymbol{\lambda}_0)F_1(\boldsymbol{\lambda}_1)[\mathrm{X}] = F_1(\boldsymbol{\lambda}_1)F_0(\boldsymbol{\lambda}_0)[\mathrm{X}]$.

**(2) Uniqueness of the transformation parameter**   The transformation parameters $\boldsymbol{\lambda}_0, ..., \boldsymbol{\lambda}_{N-1}$ are uniquely determined from X and Y.

**(3) Existence of the unit element**   For all $i(i \in \{0, ...N-1\})$, there exists a transformation parameter $\boldsymbol{\lambda}_{I,i}$ $(i \in \{0, ...N-1\})$ such that $F_i(\boldsymbol{\lambda}_{I,i})$ is an identity transformation.

## 2.2 SCENE REPRESENTATION

Scene representation is described through a relationship between two scenes $\mathrm{S}_x$ and $\mathrm{S}_y$. The relationship between $\mathrm{S}_x$ and $\mathrm{S}_y$ is formulated using $K$ transformations as follows:

$$\mathrm{S}_y = \mathrm{O}_0\mathrm{O}_1...\mathrm{O}_{K-1}\mathrm{S}_x. \tag{2}$$

In this equation, when $\mathrm{S}_x$ is a unit element, $\mathrm{O}_0, ..., \mathrm{O}_{K-1}$ are transformations that add single objects to $\mathrm{S}_x$ and represent each object in $\mathrm{S}_y$.

We investigated the algebraic conditions that are required in this equation in existing methods of scene representation learning (Burgess et al., 2019; Greff et al., 2019; Crawford & Pineau, 2019; Engelcke et al., 2019; 2021; Emami et al., 2021; Lin et al., 2020; Vikström & Ilin, 2022; Jia et al., 2022; Seitzer et al., 2023; Singh et al., 2022; Jiang et al., 2023). We extracted common algebraic conditions among all these existing methods and the results are as follows:

**(1)** The input image is decoded either by summing the component images or by a permutation-invariant process on latent vectors. These two decoding processes satisfy commutativity.

**(2)** When the input image is segmented into component images, existing methods use an auto-encoding process and uniquely bound the component image to only one latent vector that most accurately reconstructs this component image. Then a set of latent vectors is uniquely determined from the input image. Thus, if the latent vector is uniquely determined from the component image, the component image is uniquely determined from the input image.

**(3)** Existing methods can represent component images that do not affect decoding, such as component images with all 0 pixel values.

These common algebraic conditions correspond to the three conditions in algebraic independence. Therefore, we conclude that it is suggested that the necessary condition for scene representation learning is algebraic independence, which is the same regarding object representation learning. In Equation 2, generally, the transformation can be taken as an additive operation and scene $\mathrm{S}_x$ as $\mathbf{0}$, the unit element of additive operation. Then, Equation 2 is rewritten as follows:

$$\mathrm{S}_y = \mathrm{O}_0 + \mathrm{O}_1 + ... + \mathrm{O}_{K-1}. \tag{3}$$

## 3 EXPERIMENTS

### 3.1 NEURAL NETWORK MODELS

#### 3.1.1 SCENE REPRESENTATION LEARNING

We formulate scene representation learning as the segmentation of input image $Y$ into $K$ component images $Y_0, ..., Y_{K-1}$. The component image $Y_i(i \in \{0, ..., K-1\})$ is obtained by multiplying input image $Y$ by a segmentation mask $\mathbf{m}_i$. The segmentation masks $\mathbf{m}_0, ..., \mathbf{m}_{K-1}$ are obtained from a segmentation network $G_{seg}$, $(\mathbf{m}_0, ..., \mathbf{m}_{K-1}) = G_{seg}(Y)$. The size of input image $Y$ and component image $Y_i$ is $C \times H \times W$. The size of segmentation mask $\mathbf{m}_i$ is $1 \times H \times W$. The segmentation masks are constrained to have a range of values $[0, 1]$ and to satisfy $\sum_{i=0}^{K-1} \mathbf{m}_i = \mathbf{1}$ for each pixel. Therefore, component images $Y_0, ..., Y_{K-1}$ always satisfy $\sum_{i=0}^{K-1} Y_i = Y$.

#### 3.1.2 OBJECT REPRESENTATION LEARNING

We formulate object representation learning as transformation learning between two component images from different scenes $X_i$ (a component image of scene X) and $Y_i$ (a component image of scene Y). We briefly describe transformation learning in the study by Ohmura et al. (2023). An encoder $G_{enc}$ and bijective decoder $G_{dec}$ are introduced. The component image is represented by a set of multiple latent vectors. For simplicity, we describe the case in which the number of latent vectors is two. $G_{enc}$ encodes the component image $X_i$ into two latent vectors $\mathbf{x}_0^i, \mathbf{x}_1^i$, $(\mathbf{x}_0^i, \mathbf{x}_1^i) = G_{enc}(X_i)$. $G_{dec}$ decodes the two latent vectors $\mathbf{x}_0^i, \mathbf{x}_1^i$ into the component image $X_i$, $X_i = G_{dec}(\mathbf{x}_0^i, \mathbf{x}_1^i)$. Similarly, latent vectors $\mathbf{y}_0^i, \mathbf{y}_1^i$ are defined from the component image $Y_i$. Using $G_{enc}$ and $G_{dec}$, transformations $F_0(\boldsymbol{\lambda}_0)$ and $F_1(\boldsymbol{\lambda}_1)$ in Equation 1 are defined as $F_0(\boldsymbol{\lambda}_0)[X_i] = G_{dec}(\mathbf{y}_0^i, \mathbf{x}_1^i)$ and $F_1(\boldsymbol{\lambda}_1)[X_i] = G_{dec}(\mathbf{x}_0^i, \mathbf{y}_1^i)$.

We describe a loss function for training all NN models $G_{seg}$, $G_{enc}$, and $G_{dec}$. We formulate the loss function in the same manner as Ohmura et al. (2023):

$$\mathcal{L} = \sum_{i=0}^{K-1} ||Y_i - G_{dec}(\mathbf{y}_0^i, \mathbf{y}_1^{i\prime})|| + ||Y_i - G_{dec}(\mathbf{y}_0^{i\prime}, \mathbf{y}_1^i)|| \tag{4}$$

, where $(\mathbf{x}_0^{i\prime}, \mathbf{y}_1^{i\prime}) = G_{enc}G_{dec}(\mathbf{x}_0^i, \mathbf{y}_1^i)$ and $(\mathbf{y}_0^{i\prime}, \mathbf{x}_1^{i\prime}) = G_{enc}G_{dec}(\mathbf{y}_0^i, \mathbf{x}_1^i)$. The derivation of this formulation was provided by Ohmura et al. (2023).

For foreground objects, common transformations are assumed, such as color and position transformations. However, for the background, there is no such transformation. In this study, we do not consider foreground objects and the background in the same formulation because the applicable transformations are different for foreground objects and the background.

#### 3.1.3 ALGEBRAIC INDEPENDENCE

We constructed our NN models so that both scene representation learning and object representation learning satisfy algebraic independence. For scene representation learning, the summation of the component images satisfies commutativity. For the uniqueness of the component image, we use a segmentation network $G_{seg}$ that are prone to output binarized segmentation masks. As a result, the input image X is divided into multiple regions $X_0, ..., X_{K-1}$ without any overlap. And then the each region $X_i$ is uniquely bound to the single set of latent vectors $(\mathbf{x}_0^i, \mathbf{x}_1^i)$ that most accurately reconstructs $X_i$. Then a combination of latent vectors $(\mathbf{x}_0^0, \mathbf{x}_1^0), ..., (\mathbf{x}_0^{K-1}, \mathbf{x}_1^{K-1})$ is uniquely determined from the input image X. Furthermore, $(\mathbf{x}_0^i, \mathbf{x}_1^i)$ is uniquely determined from $X_i$ because $G_{dec}$ is bijective. Therefore, a combination of component images $X_0, ..., X_{K-1}$ is also uniquely determined from the input image X. For the existence of the unit element, when $X_i = \mathbf{0}$, $X_i$ does not affect the decoding of X. For object representation learning, we follow the object representation learning method based on algebraic independence in the study by Ohmura et al. (2023). Therefore, our formulation of NN models satisfies algebraic independence for both scene representation learning and object representation learning.

## 3.2 EXPERIMENTAL SETTINGS

We describe the common settings for all experiments.

**Dataset**  We conducted experiments on Multi-dSprites (Kabra et al., 2019), where the image contained multiple objects of uniform random RGB color. We set the image size to $3 \times 32 \times 32$. We normalized the pixel values from $[0, 255]$ to $[0, 1]$. We set the pixel values of the background to $0$ (black). We randomly sampled the color of each object in the range $[0.1, 1]$; the shape and angle of each object; and the position of each object under conditions that did not cause occlusion. We describe the settings for the size and number of objects individually in each section because these settings differed for each experiment.

**Networks**  As the segmentation network $G_{seg}$, we adopted U-Net (Ronneberger et al., 2015), which is widely used in image segmentation. For the parameters and normalization settings of $G_{seg}$, we adopted the settings in the study by Burgess et al. (2019), which was a scene representation learning study. We adopted the softmax function as the activation function of the final output layer of $G_{seg}$ to satisfy $\sum_{i=0}^{K-1} \mathbf{m}_i = \mathbf{1}$ for each pixel. We adopted a different number of segmentation masks $K$ for each experiment, depending on the number of objects in scenes X and Y. We adopted exactly the same encoder $G_{enc}$ and decoder $G_{dec}$ used by Ohmura et al. (2023).

**Training setup**  We trained the model for $100,000$ steps, which was sufficient for convergence. We adopted the same optimizer, learning rate, and batch size used by Ohmura et al. (2023).

**Evaluation of scene representation**  We prepared $1,000$ test images for each experiment. We evaluated the scene representation based on the quantified segmentation performance using the adjusted Rand index (ARI) (Hubert & Arabie, 1985); the larger the values the better. When the segmentation masks $\mathbf{m}_0, ..., \mathbf{m}_{K-1}$ matched the ground-truth masks, the ARI took the maximum value of $1$. We prepared ground-truth masks of foreground objects in input images X and Y.

**Evaluation of object representation**  In Multi-dSprites, the number of attributes (color, shape, position, size, and rotation) is greater than 2, which is the number of latent vectors. Thus, the correct solution is not uniquely determined for the decomposition into latent vectors. Unlike Ohmura et al. (2023), in this study, we did not perform a quantitative evaluation that assumes the existence of a unique ground-truth decomposition. We visualized the object representation by projecting a latent space into two-dimensional space using principal component analysis (PCA) (Hotelling, 1933).

## 3.3 EXPERIMENT1: TWO OBJECTS

We evaluated the NN model based on our theory in the simplest setting. We randomly sampled the pixel size of the objects in the range $[9, 14]$ for each object. We set the number of segmentation masks to 2 according to the number of objects.

Figure 1 shows an example of the resulting image after training. After 100 steps of training, single objects are not appropriately segmented. After $1,800$ steps of training, the single objects are appropriately segmented, but the learning of the transformations has not been completed. Finally, learning of the transformations also converges.

For scene representation learning, because the pixel value of the background was 0 (black), the segmentation masks did not need to be optimized for areas that did not belong to the object, which resulted in a gray mask for the background. For foreground objects, the pixel values of the segmentation masks in the region corresponding to each object converged to 1. For object representation learning, images after one transformation $F_0(\boldsymbol{\lambda}_0)[X_i]$ and $F_1(\boldsymbol{\lambda}_1)[X_i]$ showed that the latent vector $\mathbf{x}_0^i$ mainly represented color, and $\mathbf{x}_1^i$ mainly represented shape and position.

The proposed model correctly segmented single objects from input images X and Y to component images $X_0$, $X_1$, $Y_0$, and $Y_1$. The index of segmentation performance (ARI) was 0.99, which was almost the maximum value of 1, and demonstrated quantitatively that appropriate segmentation was performed. Thus, the results showed that the NN models based on our theory can appropriately perform scene representation learning.

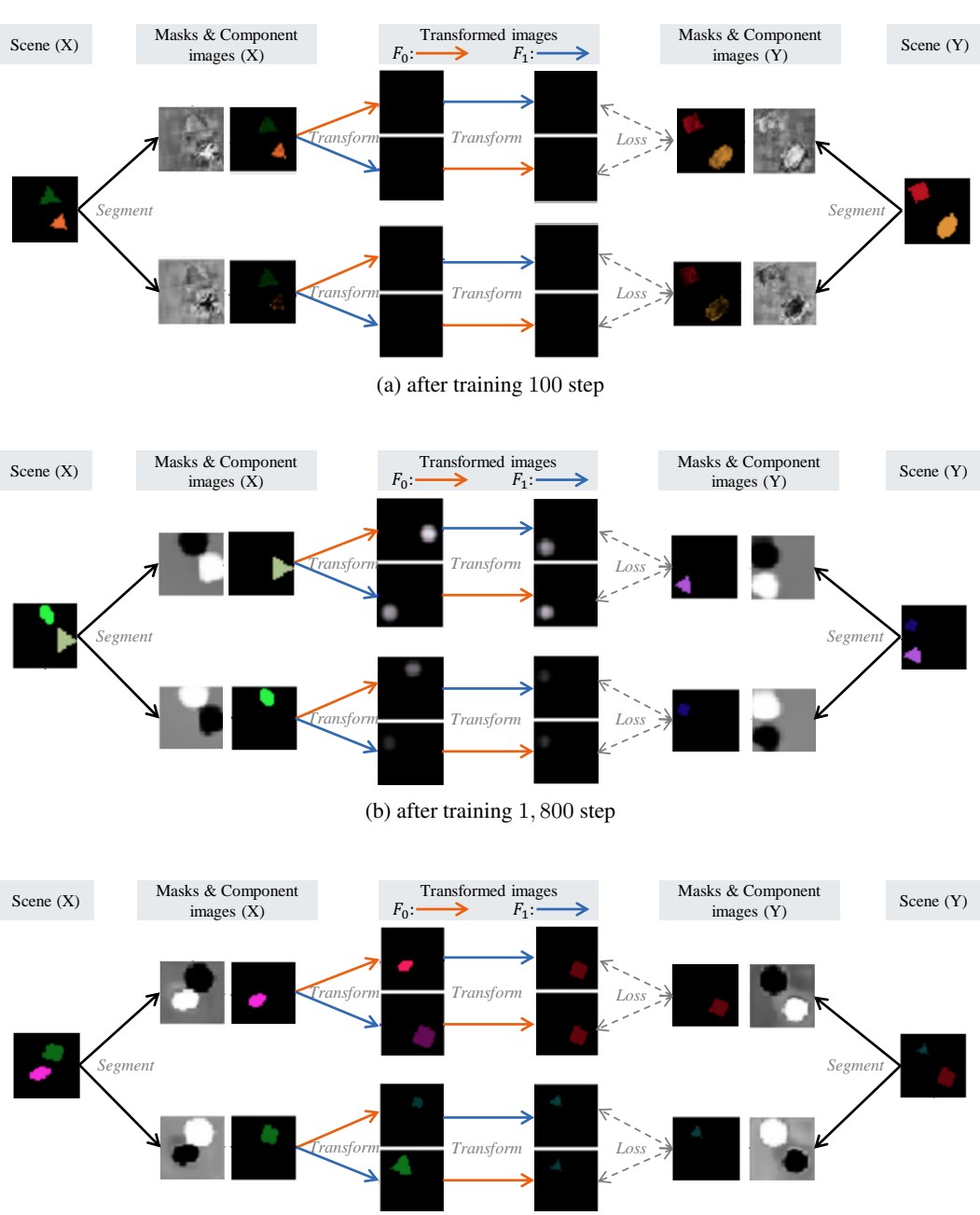

Figure 1: Result images in Experiment 1 (the number of objects in X and Y is 2) in Section 3.3. The learning progress is shown as 100 steps, 1,800 steps, and at the end of learning. From left to right: a scene containing multiple objects X, segmentation masks $\mathbf{m}_{i,x}(i \in \{0, 1\})$, component images $\mathbf{X}_i$, images after one transformation $F_0(\boldsymbol{\lambda}_0)[\mathbf{X}_i]$ and $F_1(\boldsymbol{\lambda}_1)[\mathbf{X}_i]$, images after two transformations $F_0(\boldsymbol{\lambda}_0)F_1(\boldsymbol{\lambda}_1)[\mathbf{X}_i]$ and $F_1(\boldsymbol{\lambda}_1)F_0(\boldsymbol{\lambda}_0)[\mathbf{X}_i]$, component images $\mathbf{Y}_i$, segmentation masks $\mathbf{m}_{i,y}$, and a scene Y. After 100 steps of training, single objects are not appropriately segmented. After 1,800 steps of training, the single objects are appropriately segmented, but the learning of the transformations has not been completed. Finally, learning of the transformations also converges. In this example, the 0th transformation $F_0$ mainly transformed the color, and the 1st transformation $F_1$ mainly transformed the shape and position.

Figure 2 shows the result of projecting the latent vectors $\mathbf{x}_0^i$ and $\mathbf{x}_1^i$ into points in respective two-dimensional space using PCA. The color of each point in the two-dimensional spaces indicates the color of a single object in the component image $\mathrm{X}_i$. In the 0th space for $\mathbf{x}_0^i$, the color distance is mainly reflected, but not in the 1st space for $\mathbf{x}_1^i$.

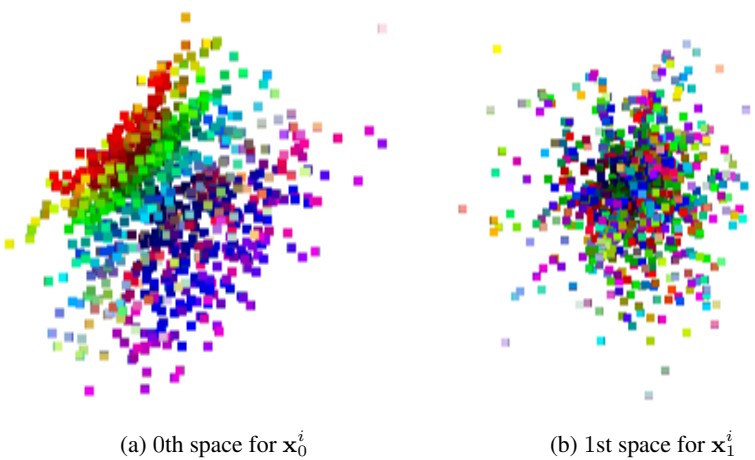

(a) 0th space for $\mathbf{x}_0^i$          (b) 1st space for $\mathbf{x}_1^i$

Figure 2: Result of projecting the latent vectors $\mathbf{x}_0^i$ and $\mathbf{x}_1^i$ into points in respective two-dimensional space using PCA. The color of each point in the two-dimensional spaces indicates the color of a single object in the component image $\mathrm{X}_i$. Contribution rate: $73.8\%$ in 0th space, $61.9\%$ in 1st space. In this example, in the 0th space for $\mathbf{x}_0^i$, the color distance is mainly reflected, but not in the 1st space for $\mathbf{x}_1^i$.

### 3.4 EXPERIMENT2: DIFFERENT NUMBER OF OBJECTS

We evaluated the proposed model in settings where the number of objects was more than 2 and the number of objects could be different in X and Y. We set the pixel size of the objects to 7. We randomly sampled the number of objects in the range $[2, 4]$. We set the number of segmentation masks to 4 according to the maximum number of objects.

Figure 3 shows an example of the resulting image after training. Images after one transformation $F_0(\boldsymbol{\lambda}_0)[\mathrm{X}_i]$ and $F_1(\boldsymbol{\lambda}_1)[\mathrm{X}_i]$ showed that the latent vector $\mathbf{x}_1^i$ mainly represented color, and $\mathbf{x}_0^i$ mainly represented shape and position.

Even if the number of objects was different, the proposed model correctly segmented single objects from input images X and Y into component images $\mathrm{X}_i$ and $\mathrm{Y}_i$ ($i \in \{0, ..., 3\}$). The index of segmentation performance (ARI) was $0.98$, which was almost the maximum value of $1$, and demonstrated quantitatively that appropriate segmentation was performed.

## 4 CONCLUSION

We proposed a unified theory of scene representation learning and object representation learning. Previous methods of scene representation learning represented a single object as a latent vector and reconstructed a scene by a decoding process that satisfied commutativity. Conventional methods of object representation learning have been based on stochastic independence, which constrains between scalar variables. Therefore, the relationship between scene representation learning and object representation learning has been unclear. We first regarded object representations as transformations between objects and scene representations as transformations between scenes. Then, we focused on the algebraic condition between transformations, and found that scene representation learning and object representation learning can be explained by a common mathematical structure - algebraic independence.

The goal of this study is to define the minimum conditions for representation learning, and we show the necessary conditions for optimal representation. Therefore, it is essential to learn better represen-

tation. For example, the number of functions for representation (in the case of scene representation, the number of objects. e.g., whether to consider a hand as five fingers or one hand) could be optimized. However, the problem is still open because the optimal representation is not static and often context-dependent.

When decomposing the attributes of a single object in our representation learning method, we often fall into suboptimal results on realistic datasets (e.g., splitting "color and shape" into "red and blue" and "shape and green"). For now, we use normalization methods that do not compromise algebraic independence, such as weight normalization (Salimans & Kingma, 2016), but they are not sufficiently effective yet. The problem of unsupervised decomposition into vectors has not been addressed nearly at all, so there are many open problems regarding techniques for optimal decomposition.

Our formulation has been based on the assumption of transformations between relatively similar objects, such as between alphabets and between polygons. Therefore, we assumed that common transformations can be applied to objects. However, with respect to background and foreground objects, the background is very different from the foreground, so applying the same transformations as those between foreground objects does not work well. Therefore, it is an open problem how transformations such as those between background and foreground objects should be learned.

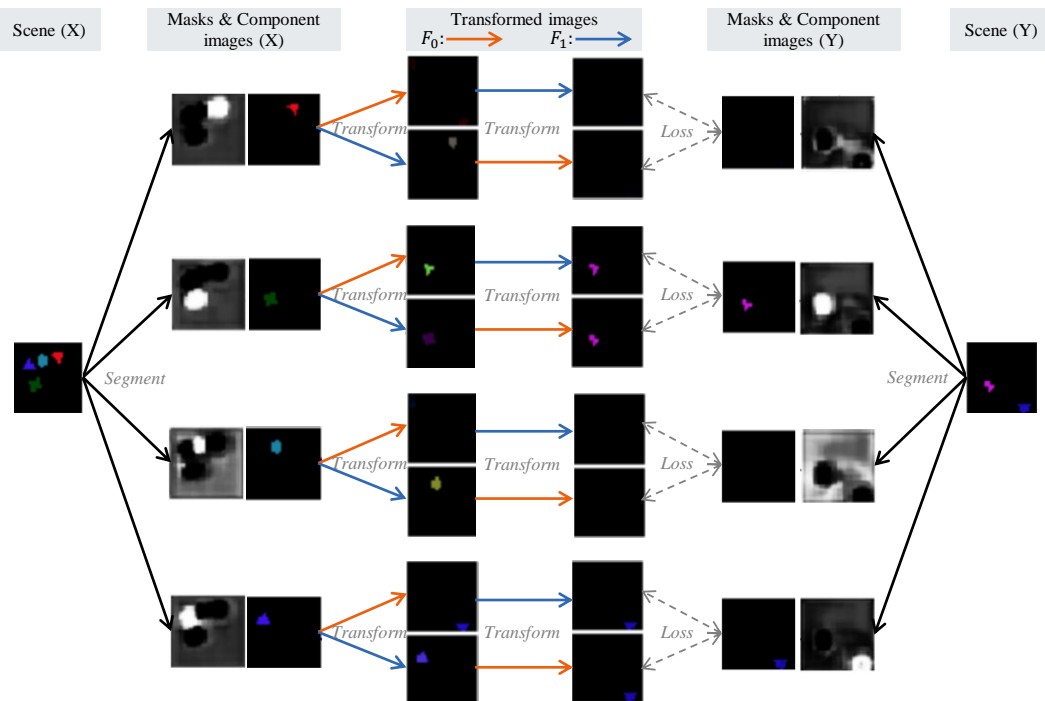

Figure 3: Result images in Experiment 2 (the number of objects in X and Y could differ) in Section 3.4. In this example, the number of objects in X was four and the number of objects in Y was two. The proposed model correctly segmented a single object from input images X and Y into component images $X_i$ and $Y_i$ ($i \in \{0, ..., 3\}$). In this example, the 0th transformation $F_0$ mainly transformed the shape and position, and the 1st transformation $F_1$ mainly transformed the color.

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
