# OpenReview forum: "A unified theory of scene representation learning and object representation learning"
_ICLR.cc/2024/Conference — Submitted to ICLR 2024_

### Official Review · Reviewer_4wri · 2023-10-30

**Soundness:** 1 poor
**Presentation:** 1 poor
**Contribution:** 1 poor
**Rating:** 3
**Confidence:** 3

**Summary:**

The paper attempts to present an approach to scene and object representation learning. Objects are decomposed into attributes and transformations between objects are learned using a neural network to satisfy algebraic independence between the attributes. A similar strategy is used to decompose scenes into objects and learn a transformation between scenes in the latent space. The authors claim the neural network used to learn these transformations accomplishes a form of representation learning.

**Strengths:**

Difficult to identify any contributions.

**Weaknesses:**

- The entire work seems like a trivial addendum to the cited work of Ohmura et. al. (2023) (which itself is a non peer reviewed paper on arXiv). Unfortunately, I was not able to identify significant extensions and significantly new experiments relative to Ohmura et. al. (2023).
- The paper is poorly written:
    - Vague explanations of technical concepts.
    - Sentences are repeated verbatim in multiple places. E.G. abstract and introduction are word-for-word almost the same.
    - Paragraph 3, line 2 seems to be copied from Ohmura et. al. (2023) page 6, line 1.
    - The authors refer to other relevant work, but contributions relative to these references are poorly motivated and poorly explained. I was not able to find explanations how and why the other works relate to the proposed work.
- I had difficulties in understanding the experiments, due to confusing setups and evaluation metrics.
- Figs 1 and 2 do not explain the setup adequately; details missing.

**Questions:**

- Motivation and utility of this approach are poorly presented. How can we use the proposed neural networks for downstream tasks?
- How would this approach handle real world images that contain complex scenes with overlapping and occluded objects?

---

### Official Review · Reviewer_cTci · 2023-10-31

**Soundness:** 3 good
**Presentation:** 2 fair
**Contribution:** 2 fair
**Rating:** 3
**Confidence:** 3

**Summary:**

The paper proposed a unified theory to explain both scene representation learning and object representation learning based on algebraic independence and validated the theory in Multi-dSprites dataset.

**Strengths:**

The algebraic independence between object representation learning and scene representation learning that this paper focuses on is worth investigating.

**Weaknesses:**

The novelty of this work seems incremental, compared to Ohmura et al. (2023), it seems that the ALGEBRAIC INDEPENDENCE is straightforwardly implemented to scene and object representations. The novelty of this paper seems not that strong.

The writing needs to be enhanced. There are some repetition in the ABSTRACT and INTRODUCTION sections, and the main contribution of the INTRODUCTION section of the paper is not very clear and concise. For example, the authors emphasized that they proposed a unified framework to explain scene and object representation learning, however, it seems difficult to follow that how  scene and object representation learning be explained.

Regarding the UNIFIED THEORY section, the Object Representation Learning Description is a quotation from existing work[1] and lacks novelty. Overall, the idea of the proposed unified explanation framework is relatively simple and straightforward.
[1] Yoshiyuki Ohmura, Wataru Shimaya, and Yasuo Kuniyoshi. An algebraic theory to discriminate qualia in the brain. arXiv preprint arXiv:2306.00239, 2023.

**Questions:**

In addition to the above weakness, I also have concerns about the experimental results.
This paper consider the segmentation as the evaluation task, and uses a very traditional measurement adjusted Rand index (ARI) in evaluation. However, the widely used segmentation measurement of accuracy, mIoU are not considered. And there is no comparison to the related works in experiments, making it difficult to evaluate the advances of this paper.

---

### Official Review · Reviewer_wsoP · 2023-10-31

**Soundness:** 1 poor
**Presentation:** 2 fair
**Contribution:** 1 poor
**Rating:** 3
**Confidence:** 5

**Summary:**

This paper proposes a unified theory approach for learning object and scene representations. The authors draw inspiration from Ohmura et al. and propose an algebraic approach to object-scene representation learning. The authors claim to have unified the learning of both object and scene representations, particularly for scenes, which had not been achieved previously. They perform a quantitative analysis on multi-colored dsprites using the ARI score to evaluate scene representation.

**Strengths:**

1. The authors present a unified approach to learning both object and scene representations through algebraic methods.

2. The authors achieve high ARI scores of 0.99 and 0.98 on two different experiments, with the number of objects varying, indicating a strong performance in scene representation.

**Weaknesses:**

1. The paper lacks novelty, as most of the work is derived from Ohmura et al.'s work.

2. The theory section is largely based on the inspired work, with limited original content.

3. Equation 4, the loss function, is also a derivative of prior work.

4. The experimental section is limited, with only two settings (two objects and different number of objects) and no comparison to other baselines.

1. The authors do not provide an evaluation of object representation.

2. The caption for Figure 2 is unclear and does not effectively convey the author's intentions.

3. There is no other ablation provided on the independence of the approach.

4. The authors do not explain the O transformation in the two experiments.

5. It is difficult to understand the approach's benefits since there is no direct comparison with the SOTA methods. Further, the authors modified the multi-dsprites dataset (so no occlusions are present), making an even harder comparison with previous approaches. We suggest the authors train SOTA scene/object representation methods for the same setup.

6. Another problem with evaluation is that the tested method was tested on only a simplified version of the multi-dsprites dataset. We encourage the authors to perform additional experiments on other datasets like CLEVR (a more challenging dataset than multi-sprites), and Tetrominoes.

7. If the proposed method cannot handle cases where objects are partially occluded, the authors should clearly mention this in their paper as a limitation of their approach.

8. The paper used segmentation accuracy as the quantitative metric. However, as the representations are derived from segmentation masks, segmentation accuracy cannot be used as a proper metric for evaluating representations. (Disentanglement metrics and other downstream tasks are needed)

9. There are multiple attributes in the dataset while the paper only involves 2 sets of latent vectors and 2 transformations, which means the proposed method is limited in generalizing to real scenario.

10. How to determine the correspondence between objects among the two scenes (how to determine the object pairs for transformation) is not clear.

**Questions:**

View the section above.

---

### Official Review · Reviewer_rWCq · 2023-11-01

**Soundness:** 2 fair
**Presentation:** 2 fair
**Contribution:** 2 fair
**Rating:** 3
**Confidence:** 4

**Summary:**

A recently proposed theory for object representation learning is used to develop a unifies theory of object and scene representation learning.
Experimental results using two simple datasets are presented.

**Strengths:**

Extension of a recent ":theory "of object representation learning to integrate scene representation learning and object representation learning.

**Weaknesses:**

The proposed theory should be validated using more challenging datasets. The proposed theory does not account for background. The modeled transformations are too simplistic. How will this work when one object is occluded by another or when 3D translational and rotational transformations are considered?

**Questions:**

Generalize the proposed theory to more challenging datasets and transformations.

---

### Meta-Review · Area_Chair_gVJs · 2023-12-09

**Metareview:**

This paper proposes an algebraic theory of object and scene representations. The basic idea is to model objects as obtained from transformations of the attributes from a canonical object. The paper constructs similar transformations between scenes. The reviewers have pointed out the technical content of this paper is a marginal extension of the existing (unpublished) work of Ohmura et. al. (2023). They also suggested that this paper will strengthened by having experiments on more realistic problems beyond the toy experiments currently in the manuscript. The authors have not provided a rebuttal to these comments.

**Justification For Why Not Higher Score:**

N/A

**Justification For Why Not Lower Score:**

N/A

---

### Decision · Program_Chairs · 2024-01-16

Reject